# Influence of Dose, Particle Size and Concentration on Dermal Penetration Efficacy of Curcumin

**DOI:** 10.3390/pharmaceutics15112645

**Published:** 2023-11-20

**Authors:** Em-on Chaiprateep, Sabrina Wiemann, Ralph W. Eckert, Christian Raab, Soma Sengupta, Cornelia M. Keck

**Affiliations:** 1Department of Pharmaceutics and Biopharmaceutics, Philipps-Universität Marburg, Robert-Koch-Str. 4, 35037 Marburg, Germany; emon_c@rmutt.ac.th (E.-o.C.); sengupts@pharmazie.uni-marburg.de (S.S.); 2Faculty of Integrative Medicine, Rajamangala University of Technology Thanyaburi (RMUTT), Thanyaburi 12130, Thailand

**Keywords:** nanocrystals, curcumin, aqueous meniscus, dermal penetration, ex-vivo model, porcine skin

## Abstract

The influence of size, particle concentration and applied dose (finite vs. infinite dose) on the dermal penetration efficacy of curcumin was investigated in this study. For this, curcumin suspensions with different particle sizes (approx. 20 µm and approx. 250 nm) were produced in different concentrations (0.625–5% (*w*/*w*)). The dermal penetration efficacy was determined semi-quantitatively on the ex vivo porcine ear model. The results demonstrated that the presence of particles increases the dermal penetration efficacy of the active compounds being dissolved in the water phase of the formulation. The reason for this is the formation of an aqueous meniscus that develops between particles and skin due to the partial evaporation of water from the vehicle after topical application. The aqueous meniscus contains dissolved active ingredients, and therefore creates a small local spot with a locally high concentration gradient that leads to improved dermal penetration. The increase in penetration efficacy depends on the number of particles in the vehicle, i.e., higher numbers of particles and longer contact times lead to higher penetration efficacy. Therefore, nanocrystals with a high particle concentration were found to be the most suitable formulation principle for efficient and deep dermal penetration of poorly water-soluble active ingredients.

## 1. Introduction

Dermal drug delivery aims to transport active pharmaceutical ingredients (APIs) into or through the skin. Today, various formulation approaches are available for this. Classical approaches include, for example, creams, ointments, lotions, gels or pastes. However, if the use of classical formulations is not sufficient, modern and sophisticated drug delivery systems are required. Besides modern vehicles that include, e.g., foams, film forming formulation or patches with and without microneedles [1,2,3], pharmaceutical nanocarriers are widely used to improve the dermal and transdermal penetration of APIs. Examples of pharmaceutical (and cosmeceutical) nanocarriers include nanoemulsions, lipid nanoparticles (SLN, NLC and smartLipids), liposomes, cochleates, niosomes or drug nanocrystals [4,5,6,7,8].

The drug nanocrystals are composed of 100% APIs, possess sizes < 1 µm and are well recognized for their improved dissolution rate and increased kinetic solubility when compared to larger sized materials. Due to this, the formulation principle is already often exploited to improve the oral bioavailability of poorly soluble APIs [9,10,11,12]. However, the dermal drug delivery of poorly soluble APIs can also be fostered with drug nanocrystals [13,14,15,16,17,18,19,20,21,22]. Thereby, the improved kinetic solubility (c_s_) of the nanocrystals is considered to be the major parameter that promotes increased dermal penetration of the API. The increased c_s_ results in an increased concentration gradient (dc/dt) of dissolved API between formulation and skin, and thus—based on Fick’s law—results in an increased passive diffusion of the API molecules into or through the skin.

The above-mentioned mechanism is based on current textbook knowledge and relies on the consideration that particles in a vehicle will dissolve until they reach their saturation solubility in the surrounding medium. In the case of topical formulations, this surrounding medium is most likely a liquid or a semi-solid vehicle (gel, cream, ointment). After topical application, the amount of dissolved APIs in the vehicle is considered to be the driving force for the passive dermal penetration of the APIs, where high amounts of dissolved API in the vehicle are believed to create a high dc/dt between the formulation and skin. Thus leading to good passive dermal penetration of the APIs and vice versa.

In order to maintain a high dc/dt between vehicle and skin, excess APIs should be added to the formulations in the form of particles. This enables dissolved API molecules, that penetrate from the vehicle into the skin, to be immediately replaced by APIs that dissolve from the particles. Thus, restoring the c_s_ of the APIs in the vehicle and maintaining a constantly high dc/dt between vehicle and skin which enables a constant and long-lasting diffusion of the API into and through the skin [23,24,25,26].

Based on this current state-of-the-art theory, the addition of undissolved API particles is only necessary to provide enough APIs to restore and maintain the c_s_ of the APIs within the vehicle. Consequently, it can be considered that the addition of a too high surplus of API particles would not create any beneficial effect. Interestingly, an opposite observation was made by Vidlářová et al. in 2016 [22]. In this study, the authors produced curcumin nanocrystals. The nanocrystals were diluted, and different curcumin concentrations were applied to porcine skin. All formulations contained curcumin in excess in nanoparticulate form. The authors demonstrated an effective dermal penetration of the poorly soluble curcumin from the nanocrystals. However, they also showed that the application of higher concentrations of curcumin nanocrystals resulted in a more pronounced diffusion of curcumin into the skin. Based on the results it was suggested that the nanocrystals adhere to the skin where they create a high local concentration gradient which results in a high local penetration of the APIs. The proposed theory was further substantiated by recent studies [27,28,29].

Besides the confirmation that higher particle concentrations and an increased amount of particles being adhered to the skin can increase the dermal penetration of APIs, it was also found that the stratum corneum (SC) thickness was significantly increased with an increased particle concentration [29]. The effect was observed for large sized bulk material and for nanocrystals and was explained by the formation of an aqueous meniscus between the particles and skin. The aqueous meniscus is considered to connect the particles to the skin via capillary forces. In addition, it leads to a local swelling of the SC, which will—besides a high local concentration gradient—further promote the local penetration of the API (Figure 1).

Consequently, it can now be hypothesised that passive dermal penetration from particle-containing topical formulations is the sum of two scenarios. The first scenario is the passive diffusion of the dissolved APIs from the vehicle, and the second scenario is the local passive penetration of the APIs from the meniscus. The theory that particles adhere to the skin via the formation of a liquid meniscus which is created upon the drying of the vehicle, was supported by the measurement of the SC thickness of non-treated skin and of skin treated with large-sized bulk material and curcumin nanocrystals. In this study, each formulation (bulk and nano) was tested with two different curcumin particle concentrations. In addition, the formation of an aqueous meniscus between a spherical particle and a surface was demonstrated by light microscopy [29]. However, a detailed investigation that allows for a detailed and mechanistic understanding of the proposed effect is not yet available. Therefore, the aim of this study was to investigate the influence of particle concentration on SC thickness and passive dermal penetration in more detail.

For this, curcumin nanocrystals and curcumin bulk suspensions were produced and characterized as described previously [29]. The formulations were diluted to obtain formulations with different particle concentrations (Table 1). Subsequently, the dermal penetration of curcumin from the different formulations was determined by using an ex vivo pig ear model. The SC thickness was also determined and compared to untreated skin and to skin treated with the particle free vehicle (surfactant solution). In addition, to investigate the effect of the dose (volume of formulation applied to the skin), each formulation was applied in high (infinite) and low (finite) doses (50 µL vs. 10 µL), respectively.

## 2. Materials and Methods

### 2.1. Materials

Curcumin (curcuma longa L. dry extract (purity: 80% curcumin; 95% curcuminoids)) was obtained from Receptura Apotheke (international compounding pharmacy, Cornelius-Apothekenbetriebs-OHG, Frankfurt a. M., Germany). TPGS (d-α-tocopherol polyethylene glycol 1000 succinate) (Gustav Parmentier GmbH, Frankfurt a. M., Germany) was used as the surfactant for the suspensions. Purified water served as the dispersion medium and was freshly obtained from a PURELAB^®^ Flex 2 (ELGA LabWater, Veolia Water Technologies Deutschland GmbH, Celle, Germany).

### 2.2. Methods

In the first part of the study, curcumin bulk suspensions and nanosuspensions with different concentrations of curcumin were produced and characterized regarding size, size distribution and zeta potential (c.f. Section 2.2.1 and Section 2.2.2). In the second part, the influence of the formulations on the stratum corneum thickness and the dermal penetration of curcumin was investigated by using an ex vivo porcine ear model and subsequent epifluorescence microscopy with digital image analysis (c.f. Section 2.2.3).

#### 2.2.1. Production of Curcumin Bulk and Nanosuspensions

The bulk suspensions contained 5% (*w*/*w*) curcumin and were stabilized with 1% (*w*/*w*) TPGS. The nanosuspensions were produced out of bulk suspensions by using small-scale bead milling after a previously described protocol [29]. Briefly, the bead milling was carried out in a 25 mL Erlenmeyer flask with an inserted magnetic stirring bar (Asteroid^®^ 25, 2mag AG, München, Germany) and 1.0–1.2 mm yttria stabilized zirconia grinding beads (SiLibeads^®^, Sigmund Lindner GmbH, Warmensteinach, Germany). The bead/suspension ratio was adjusted to 40/60 (*v*/*v*) and the flask was placed in an ice bath on top of a magnetic stirring plate (RCT standard, IKA^®^-Werke GmbH & Co. KG, Staufen, Germany). The nano-milling was conducted at 1500 rpm for 14 h. The temperature of the ice bath was constantly kept between 10 and 20 °C. Afterwards, the nanosuspensions were diluted to 2.5% (*w*/*w*), 1.25% (*w*/*w*) and 0.625% (*w*/*w*) curcumin with 1% (*w*/*w*) TPGS surfactant solution. The bulk suspensions were diluted in the same way to yield identical concentrations.

#### 2.2.2. Physicochemical Characterization of Curcumin Formulations

##### Size Characterization

Nanosuspensions were characterized using a combination of three different and independent sizing techniques, i.e., photon correlation spectroscopy (PCS), laser diffraction (LD) and light microscopy (LM). The bulk suspensions were characterized via LD and LM, respectively.

Photon Correlation Spectroscopy (PCS)

PCS was used to determine the hydrodynamic particle size diameter (z-average) and the polydispersity index (PdI) of the nanosuspensions by using a Zetasizer Nano ZS (Malvern Panalytical Ltd., Malvern, UK). Data analysis was carried out in general purpose mode. Since the Zetasizer has a sample-dependent measuring range, possible larger µm particles cannot be detected with this technique [30]. Nanosuspensions were therefore additionally measured with LD, to enable the detection of remaining larger particles after the milling process.

Laser Diffractometry (LD)

For the detection of potentially remaining larger µm-sized particles after the milling process, LD analysis was carried out. The volumetric particle size distribution (d(v)x [µm]) was determined via a Mastersizer 3000 (Malvern Panalytical Ltd., Malvern, UK). Mie theory was used for data analysis and the real refractive index was set to 1.87. The imaginary refractive indices used were different for the blue light (470 nm) and the red light (632.8 nm) laser beam, due to the fact that the UV/Vis spectrum of curcumin shows a slight absorption at a wavelength of 470 nm and no absorption at 632.8 nm. Thus, the imaginary refractive indices were set to 0.1 (blue light) and 0.01 (red light).

Light Microscopy (LM)

LM, using an Olympus BX53 light microscope (Olympus Cooperation, Tokyo, Japan), which was equipped with an Olympus SC50 CMOS color camera (Olympus soft imaging solutions GmbH, Münster, Germany) was used to check for particle agglomeration.

##### Zeta Potential (ZP) Analysis

The zeta potential (ZP) of the curcumin nanosuspensions was determined by measuring the electrophoretic mobility (EM) via laser Doppler anemometry. The ZP was calculated from the EM obtained using the Helmholtz–Smoluchowski equation. The measurements were carried out with a Zetasizer Nano ZS (Malvern Panalytical Ltd., Malvern, UK) at 20 °C. Subsequently, prior to the analysis, samples were either dispersed in their original dispersion medium (1% (*w*/*w*) TPGS), or in water adjusted to a conductivity of 50 µS/cm at 20 °C with NaCl solution [31].

#### 2.2.3. Determination of Stratum Corneum Thickness and Dermal Penetration Efficacy

In order to investigate the influence of the formulations on the SC thickness and the dermal penetration behaviour of curcumin, the ex vivo model using porcine ears was used [28,29]. For this purpose, fresh pig ears were obtained from a local slaughterhouse and used within a few hours after slaughter.

Firstly, the ears were washed with lukewarm water (with temperature of approx. 23–25 °C) and gently dried with a paper towel in dabbing movements without rubbing. Secondly, examination areas of 2 × 2 cm (corresponding to an area of 4 cm^2^) on the dorsal side of the ears were determined, whereby it was ensured that these skin areas were intact and free of injuries. The hairs in these areas were carefully trimmed to a length of about 1–3 mm with scissors.

Subsequently, the formulations (Table 1) were applied to each examination area and carefully distributed with a pipette tip without massage. The amounts applied to the skin were selected to represent either a finite (<10 µL/cm^2^) or infinite dose (≥10 µL/cm^2^) setup, respectively [32]. The finite dose setup was obtained by applying 10 µL of the formulation to the 4 cm^2^ skin areas, and the infinite dose setup was obtained by applying 50 µL of the formulation to the 4 cm^2^ skin areas. This corresponded to doses of 2.5 µL/cm^2^ skin (finite dose) and 12.5 µL/cm^2^ skin (infinite dose), respectively (Table 1). Untreated skin areas served as controls.

After 4 h of incubation time at 32 °C, the formulations were carefully rinsed off with water. The ears were dried with a paper towel, and subsequently, punch biopsies (15 mm) were taken. The excised skin pieces were embedded in Tissue-Tek^®^ O.C.T.™ (Sakura Finetek Europe B.V., Alphen aan den Rijn, The Netherlands) and immediately frozen at −80 °C until further use. The experiments were carried out in triplicate.

In the next step, the frozen skin samples were dissected into vertical skin cuts of 20 µm thickness with a cryomicrotome (Frigocut 2700, Reichert-Jung, Nußloch, Germany). Since curcumin has a strong autofluorescence its presence in the skin, i.e., the extent of its dermal penetration, can be easily tracked via inverted epifluorescence microscopy [27,33].

For this purpose, the skin sections were analyzed with an inverted epifluorescence microscope (Olympus CKX53 equipped with an Olympus DP22 color camera, Olympus Deutschland GmbH, Hamburg, Germany). To quench the autofluorescence of the skin to a minimum, the intensity of the fluorescent light source was set to 50% and the exposure time was adjusted to 50 ms. The settings were kept constant during analysis. The selected fluorescence filter for the analysis was the DAPI HC filter block system (excitation filter: 460–500 nm, dichroic mirror: 500 nm, emission filter: from 500 nm (LP)).

##### Digital Image Analysis

With the help of digital image analysis, the penetration behavior of curcumin, derived from the different formulations, could be further examined. In this context, the mean penetration depth (MPD) of curcumin and the reaction of the stratum corneum (SC) thickness upon treatment, i.e., swelling and shrinking, were determined. The analysis was done via the ImageJ software version 1.8.0 [34,35]. By using the scale bar of the software, the penetration depth (PD) and the SC thickness (SCT) were measured on each recorded image [28,29].

As a further parameter, the mean grey value (MGV) for each formulation was determined. This value served as a semi-quantitative measurement of the total amount of the penetrated curcumin. This was carried out with a previously defined protocol [33]. An automated threshold protocol was performed with the aim of eliminating the autofluorescence of the skin. As a consequence, the intensity of the curcumin remained [33].

#### 2.2.4. Statistical Analysis

The software Microsoft Excel^®^ (Version 2021) was used to calculate the descriptive statistics. Inferential statistics were calculated with the JASP software (Version 0.13.1). The analysis included testing for normal distribution and variance homogeneity with the Shapiro–Wilk and Levene’s tests, respectively. Means of normally distributed data were compared in a Welch’s ANOVA in case of variance heterogeneity. Furthermore, Tukey’s and Games–Howell post hoc tests were conducted.

Kruskal–Wallis analysis of variance was performed for non-parametric data. Additionally, a direct comparison of some selected data was considered. In this case, the Student’s *t*-test for independent samples was used for the normally distributed data and the Wilcoxon rank-sum test was calculated for not-normally distributed data. A *p*-value < 0.05 was defined as statistically significant. Error bars in the figures represent the standard deviations.

## 3. Results and Discussion

### 3.1. Production and Characterization of Curcumin Bulk and Nanosuspensions

The curcumin bulk suspensions possessed a particle size of about 20 µm (LD data, d(v)0.5) and contained no larger particles (d(v)0.95 < 100 µm, d(v)0.99 < 150 µm, Figure 2A). The PCS particle size of the curcumin nanosuspensions was in the range of about 220 and 250 nm (Figure 2B). The size was smallest for the suspensions that contained 5% curcumin and increased with decreasing concentration. The effect is caused by a partial dissolution of the curcumin nanocrystals that occurs upon dilution of the formulations [36]. Nevertheless, the difference in size is small, and thus can be considered to have no effect on the dissolution velocity and kinetic solubility of the curcumin [37]. In addition, the zeta potential was affected by the dilution of the particles, i.e., it was smaller when the formulations contained less particles (Figure 2C). This effect is also reasonable because all formulations contained similar amounts of surfactant, but different amounts of particles. Therefore, for formulations that contained more particles, less surfactant, when compared to formulations with fewer particles, is available. This means the surfactant layer will be thinner for these formulations. As TPGS is a non-ionic surfactant, it stabilizes particles via a steric stabilization mechanism. Hence, a thicker surfactant layer can be identified by a lower zeta potential and vice versa. Therefore, the lower zeta potential observed for the formulations nicely demonstrates that the surfactant layer is slightly thinner for the formulations that contain more particles. In real life, this would mean that these formulations might be less physically stable during long-term storage than the formulations that contained fewer particles. However, in this study, the small differences in zeta potential can be neglected, because the particles were freshly prepared prior to their dermal application. The particles were applied in surfactant solution, i.e., their original dispersion medium. In this medium, the zeta potential was similar for all formulations. Therefore, the surface charge of the particle, when applied on skin, was similar for all formulations. Additionally, all other parameters that can affect the dermal penetration efficacy, e.g., viscosity, were considered to be similar for each formulation, as all formulations contained the same dispersion medium. Therefore, the effects that were observed after dermal application can be considered to be related only to their differences in size (bulk—20 µm vs. nano—220–250 nm), their applied dose (finite dose—2.5 µL/cm^2^ vs. infinite dose 12.5 µL/cm^2^) and their particle concentration (0.625%, vs. 1.25% vs. 2.5% vs. 5%).

### 3.2. Influence of Particle Concentration, Dose and Particle Size on SC Thickness

The next step of the study aimed at investigating the influence of size, dose and number of particles on the SC thickness, which is a surrogate for skin hydration, i.e., a thicker SC corresponds to a more hydrated SC and vice versa [28,29,38,39].

The thickness of non-treated skin (blank) was about 32 µm after 4 h of incubation at 32 °C (Figure 3A). The addition of low amounts of surfactant (10 µL) did not alter the SC thickness, but the application of 50 µL surfactant solution decreased the SC thickness significantly (Figure 3A), indicating that higher amounts of surfactant can impair the SC’s integrity, which can lead to an increased water loss from the skin. The increased water loss results in the drying of the skin and leads to a decreased SC thickness [38].

Application of the curcumin suspensions increased the SC thickness with increasing curcumin concentration (Figure 3B). The effect was most pronounced for the curcumin bulk suspensions and less pronounced for the curcumin nanosuspensions. The results therefore confirm the theory that curcumin particles can increase the SC thickness [29]. The differences between the different formulations can further substantiate this theory.

The stronger increase in SC thickness for the bulk material, when compared to the nanomaterial, can be explained by the fast sedimentation of the large particles (Figure 4, [29]). The large particles (>10 µm) settle quickly after dermal application, and thus come into contact with the skin quickly, where they create a locally high concentration gradient of dissolved curcumin. Over time, the water evaporates, and the remaining water forms an aqueous meniscus between particles and skin. Thus, promoting a fast onset of the swelling of the SC. Higher doses of curcumin suspension (infinite = 12.5 µL/cm^2^ vs. finite = 2.5 µL/cm^2^) provide more particles which can settle on the skin. Therefore, the number of particles that are connected to the skin is higher and consequently, the SC swelling effect is more pronounced when higher doses, i.e., more particles, are applied to the skin (Figure 4A,B).

The curcumin nanosuspensions contain particles < 1 µm which are too small to sediment on the skin (Figure 4C,D). Over time, the water evaporates from the formulation and forces the particles to come into close contact with the skin. Only after a sufficient evaporation of the water, the particles are able to form the aqueous meniscus between the particles and skin and to promote swelling of the SC. The drying of the formulation is accomplished faster with the lower doses applied (2.5 µL/cm^2^—finite dose) and will take longer for the higher doses (12.5 µL/cm^2^—infinite dose). Hence, in contrast to the bulk material, where higher doses resulted in a faster swelling of the SC, the swelling of the SC starts later for the higher doses (50 µL) of the curcumin nanomaterial (Figure 4C,D). This results in a shorter time in which particles are in close contact with the skin, and thus to a less-efficient swelling of the SC when compared to the bulk material and the low dose nanomaterial.

The results demonstrate that the degree of SC swelling is a superimposed effect. It is affected by at least two parameters, i.e., the area of skin that is connected to particles via an aqueous meniscus (A_particles_), and by the time where the aqueous meniscus can be maintained on the skin (t_meniscus_). Consequently, formulations that create both, high A_particles_ and t_meniscus_ values, can be expected to lead to the most efficient dermal and/or transdermal penetration of active compounds.

### 3.3. Influence of Particle Concentration, Dose and Particle Size on Dermal Penetration Efficacy

Based on the results obtained it can be expected that a higher amount of larger-sized particles leads to a better and deeper penetration than lower amounts of larger-sized particles. Consequently, it was expected that increasing curcumin particle concentrations will lead to increasing amounts of penetrated curcumin. In addition, the application of 50 µL (infinite dose) was expected to lead to an improved penetration of curcumin when compared to the lower dose (10 µL, finite dose). The results obtained fully confirmed these expectations (Figure 5 and Figure 6).

The effect was significant for the penetration efficacy for all particle concentrations of the bulk material (Figure 5A) and this trend was also seen for the penetration depth (Figure 5B).

Additionally, for the nanomaterials, an increase in penetration efficacy was expected with increasing curcumin particle concentration. However, in contrast to the bulk material where larger doses (volumes) led to a better penetration, the application of higher doses (infinite doses) of the nanomaterial—due to the slower sedimentation of the particles—were expected to lead to less penetration when compared to the lower doses (finite doses, cf. Figure 4). Indeed, this expected increase in dermal penetration efficacy and the penetration depth of curcumin with increasing particle concentration could be confirmed (Figure 6A,B). However, the expected increase in penetration efficacy for lower, finite doses was only seen as a slight trend, but was not significant. Data therefore indicate that the number of particles that are connected to the skin (A_particles_) is the major driving parameter that influences dermal penetration efficacy when nanomaterials are exploited for dermal application. In contrast, the applied dose (finite vs. infinite) seems to have a less pronounced effect on the penetration efficacy.

A possible reason for the less pronounced dose effect is that the nanocrystals are too small to settle to the skin. Hence, as long as enough liquid is on top of the skin, only the dispersion medium is in contact with the skin (cf. Figure 4). The dispersion medium, independent on the concentration of undissolved curcumin particles, contains equal amounts of dissolved curcumin molecules, i.e., it represents—due to the small size of the nanocrystals—a supersaturated solution of curcumin [9,11]. Hence, in contrast to the bulk material where particles settle, the dermal penetration of the dissolved active from the nanocrystals can be considered to be similar for both formulations during this time. Only after all of the liquid has disappeared, the particles can get into contact with the skin and can then create the locally high concentration gradient. The lower dose, i.e., a smaller volume applied to the skin, can be expected to evaporate faster, and therefore, the particles can connect earlier to the skin. Thus, leading to an increased penetration of the actives when compared to the higher doses.

### 3.4. Comparison of Penetration Efficacy from Bulk Material and Nanocrystals

Various literature describes the superiority of drug nanocrystals for dermal drug delivery when compared to larger sized bulk material [22]. However, only very rare data that investigate the influence of size of drug crystals on the dermal penetration are available today [40]. A systematic study that investigates not only the influence of size, but also dose and particle concentration on the dermal penetration efficacy, is not yet available and was, therefore, pursued in this study (Figure 7).

Data show that all parameters affect the dermal penetration efficacy (Figure 7A) and penetration depth (Figure 7B). The effects of particle number were discussed in the sections above. Therefore, in this part of the study, the focus is put on the influence of the particle size and applied dose. Data show that smaller particles allow for a deeper and higher penetration of the curcumin when applied in a finite dose setup (Figure 7 left, Figure 8). In contrast, they result in a lower penetration efficacy, i.e., less penetrated curcumin molecules, when applied in an infinite dose setup (Figure 7 right, Figure 8). Hence, the benefit of nanocrystals could only be observed when the formulations were applied in finite dose setup.

The reason for these opposing results is the sedimentation of the larger sized bulk particles during the experiment. The large particles, due to their size, i.e., heavy weight, sediment to the surface of the skin. The infinite dose (50 µL) contains a 5-fold higher number of particles than the finite dose (10 µL). As all particles within the suspension can be considered to settle to the skin, much higher doses are in contact with the skin in the case of the infinite dose setup (cf. Figure 4). The particles in contact with the skin create a high local concentration gradient and thus cause a higher penetration when compared to the finite dose setup. The nanocrystals, due to their small size, cannot sediment. Therefore, the dose applied is not a major parameter affecting the penetration efficacy (Figure 9).

The finite dose mimics the setup of a real application; whereas, the infinite dose setup is recommended to be used for investigating the flux of active compounds through the skin but is known to not mimic real world dermal application [41]. Data therefore suggest that the testing of the dermal penetration efficacy of particle containing formulations that can sediment during the test should only be performed in a finite dose setup and by applying quantities to the skin that mimic a real in vivo application of such formulations. This finding is in line with a previous study that revealed similar results [29].

## 4. Conclusions

Curcumin bulk formulations in micrometer size and curcumin nanocrystals in submicron size with different concentrations of particles (0.625–5% (*w*/*w*)) were produced in this study and used to determine the influence of size, particle concentration and applied dose on the dermal penetration efficacy of curcumin. The results provide evidence that the particles in a formulation can increase the dermal penetration efficacy of active compounds. The suggested mechanism is the formation of a meniscus between the particles and skin. The meniscus is composed of the formulation in which the particles are suspended. The attachment of the particle to the skin creates a high local concentration gradient which fosters the passive diffusion of the dissolved active ingredient into the skin. In addition, it was found that the meniscus leads to a local swelling of the stratum corneum, which can further increase the passive dermal diffusion of the active compounds. Major parameters that led to an efficient increase in dermal penetration were found to be (i) the area of skin being connected to particles via the meniscus (A_particles_) and (ii) the time period in which the meniscus can be maintained on the skin (t_meniscus_).

Nanocrystals with a high particle concentration were found to be the most suitable formulation principle for the efficient and deep dermal penetration of active ingredients. The results also demonstrate the sensitivity of the ex vivo porcine ear model and the need to test the dermal penetration efficacy of particle-containing formulations in finite test models. The testing in infinite dose setup can be misleading and can lead to artefacts that are created due to the sedimentation of larger particles. These artifacts will not occur in a real in vivo application where finite doses are applied to the skin.

## Figures and Tables

**Figure 1 pharmaceutics-15-02645-f001:**
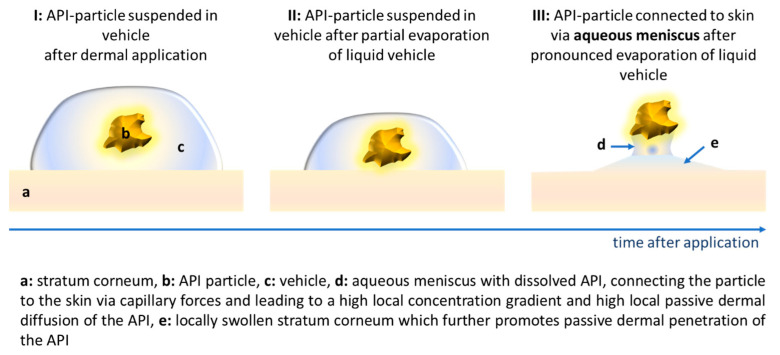
Schemes of proposed mechanism for passive dermal penetration of APIs from API-particle containing vehicles.

**Figure 2 pharmaceutics-15-02645-f002:**
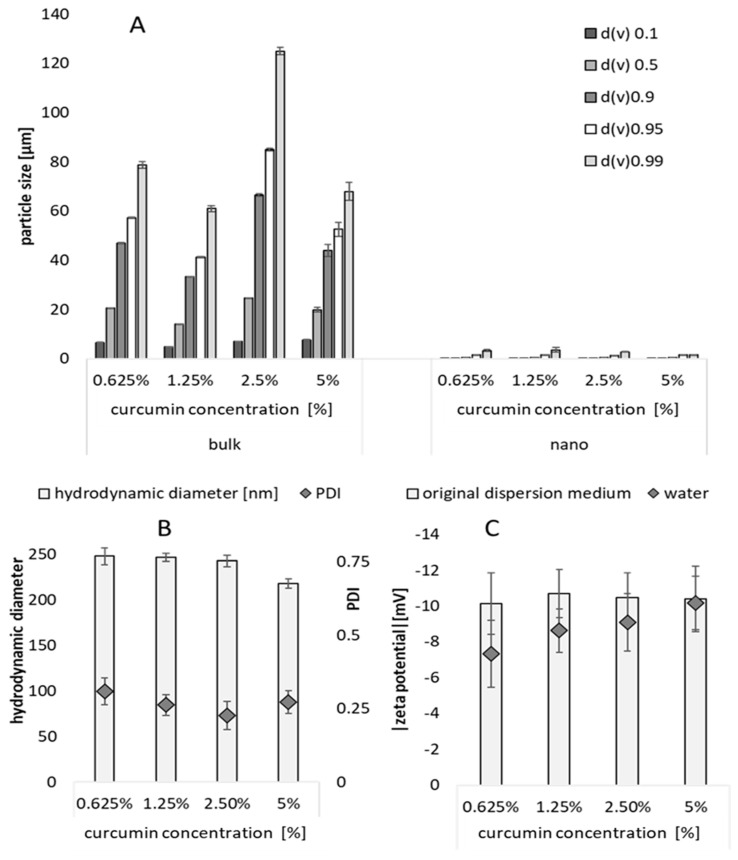
Characterization of curcumin bulk and nanosuspensions. (**A**) Particle size analysis with LD. (**B**) Particle size analysis with PCS. (**C**) Zeta potential analysis in original dispersion medium (TPGS, 1%) and in water.

**Figure 3 pharmaceutics-15-02645-f003:**
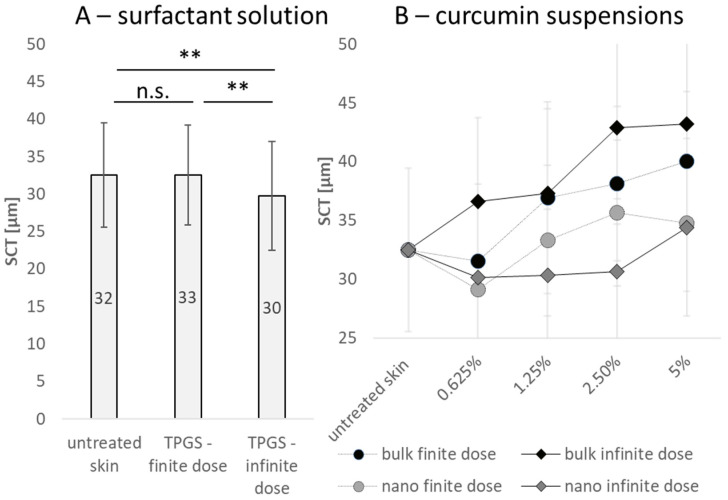
Influence of treatment on stratum corneum thickness (SCT). (**A**) Influence of surfactant (1% TPGS) and applied dose (finite ≜ 2.5 µL/cm^2^ applied dose and infinite dose ≜ 12.5 µL/cm^2^ applied dose) on SCT. (**B**) Influence of curcumin particle concentration, size and applied dose on SCT. ** *p* < 0.01, n.s.—non-significant.

**Figure 4 pharmaceutics-15-02645-f004:**
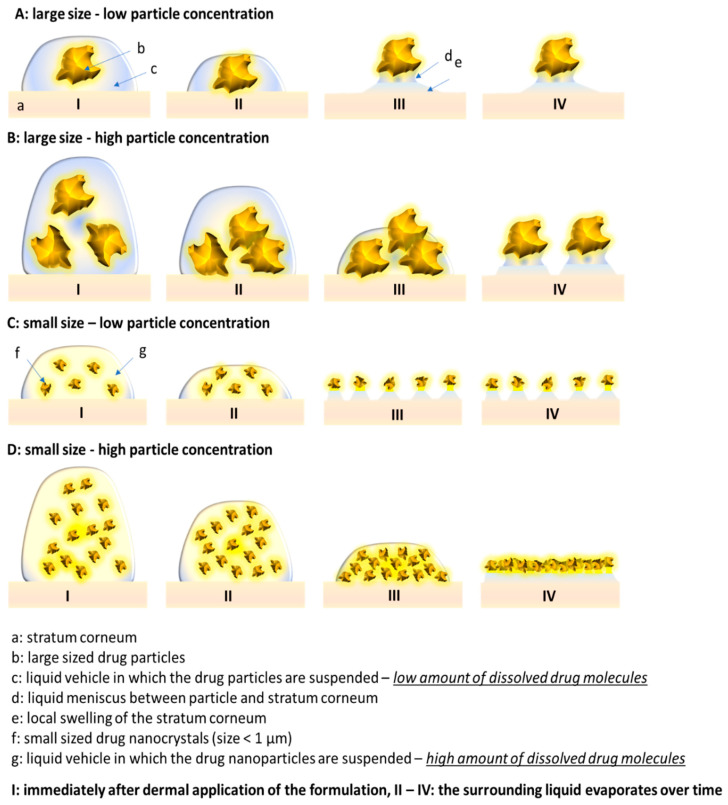
Scheme that illustrates the influence of size, dose and particle number on stratum corneum thickness (SCT). (**A**) large sized bulk material—finite dose, (**B**) large sized bulk material—infinite dose, (**C**) small sized nanosuspension—finite dose, (**D**) small sized nanosuspension—infinite dose (detailed explanations of effects—cf. text).

**Figure 5 pharmaceutics-15-02645-f005:**
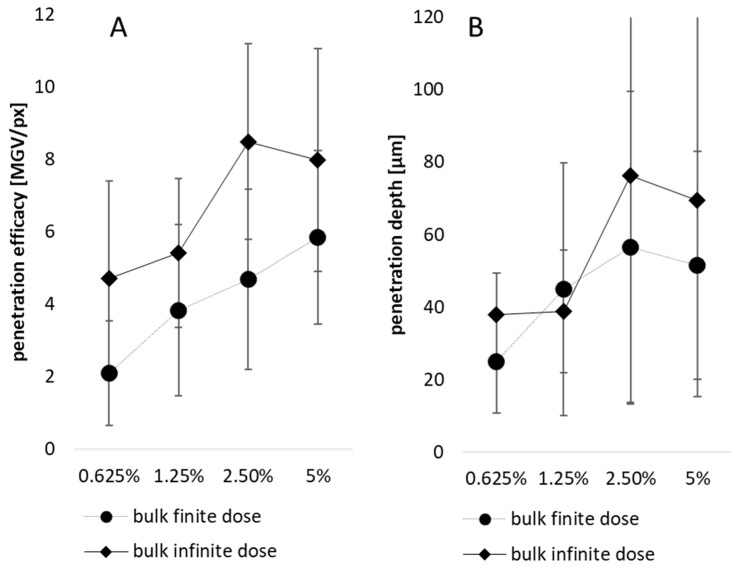
Influence of particle concentration and dose applied on (**A**) The dermal penetration efficacy and (**B**) Penetration depth—bulk material.

**Figure 6 pharmaceutics-15-02645-f006:**
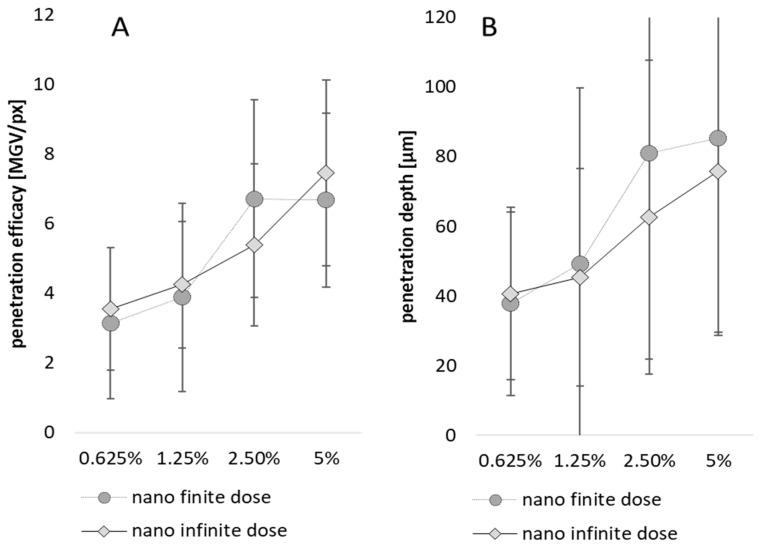
Influence of particle concentration and dose applied on (**A**) the dermal penetration efficacy and (**B**) penetration depth—nanosuspensions.

**Figure 7 pharmaceutics-15-02645-f007:**
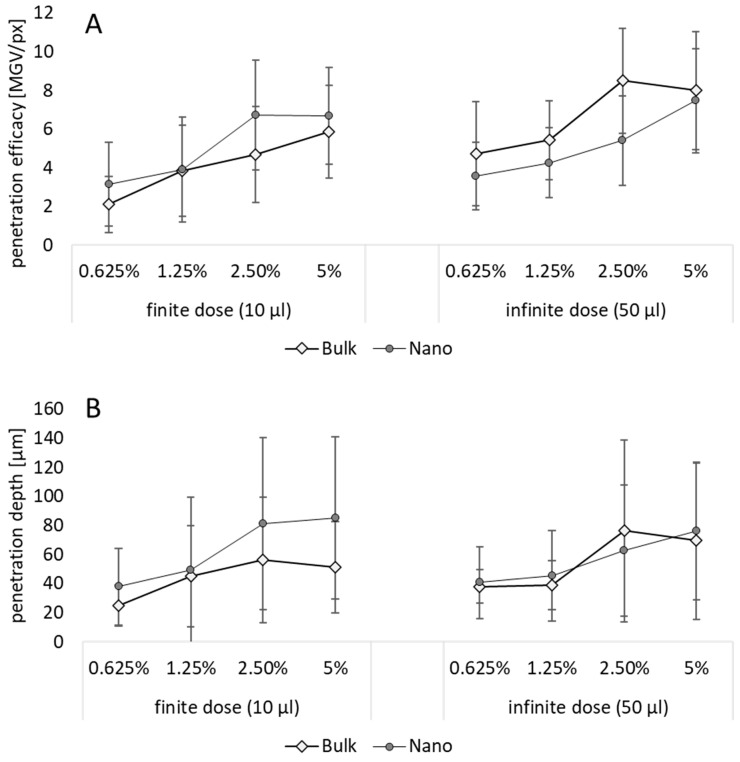
Influence of particle size, particle number and dose on dermal penetration efficacy (**A**) and penetration depth (**B**).

**Figure 8 pharmaceutics-15-02645-f008:**
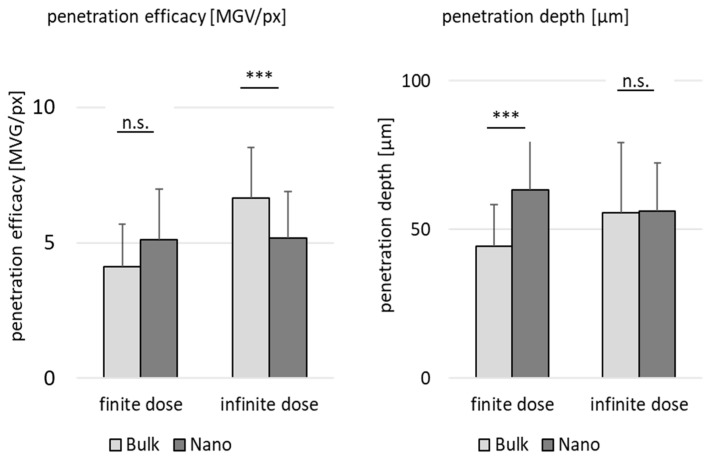
Influence of particle size and dose on dermal penetration efficacy (**left**) and penetration depth (**right**). The graph provides a direct comparison on the influence of the size (bulk vs. nano) in the dermal penetration efficacy and penetration depth in either finite dose or infinite dose setup. *** *p* < 0.001, n.s.—non-significant.

**Figure 9 pharmaceutics-15-02645-f009:**
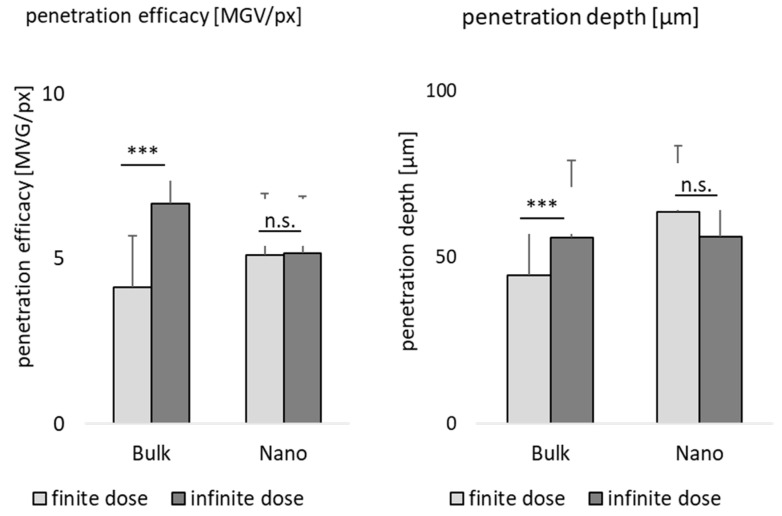
Influence of dose and particle size on dermal penetration efficacy (**left**) and penetration depth (**right**). The graph provides a direct comparison on the influence of the dose, i.e., the volume of formulation applied to the skin (finite vs. infinite), on the dermal penetration efficacy and penetration depth with either large sized bulk material (20 µm = Bulk) or nanosized curcumin crystals (220–250 nm = Nano). *** *p* < 0.001, n.s.—non-significant.

**Table 1 pharmaceutics-15-02645-t001:** Overview of formulations produced and applied on skin.

Formulation	Curcumin Concentration	Applied Dose = Volume Applied on Skin	
surfactant (1% TPGS)	0% (*w*/*w*)	finite dose = 2.5 µL/cm^2^ corresponds to a volume of 10 µL that was applied on the skin area of 2 × 2 cm (4 cm^2^)	infinite dose = 12.5 µL/cm^2^ corresponds to a volume of 50 µL that was applied on the skin area of 2 × 2 cm (4 cm^2^)
curcumin bulk suspension	0.625% (*w*/*w*)		
	1.25% (*w*/*w*)		
	2.5% (*w*/*w*)		
	5.0% (*w*/*w*)		
curcumin nano suspension	0.625% (*w*/*w*)		
	1.25% (*w*/*w*)		
	2.5% (*w*/*w*)		
	5.0% (*w*/*w*)		

## Data Availability

Data sharing is not applicable to this article.

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
