# Peer review of "Influence of Dose, Particle Size and Concentration on Dermal Penetration Efficacy of Curcumin"

_pharmaceutics, 2023, doi:10.3390/pharmaceutics15112645_

Round 1

Reviewer 1 Report

Comments and Suggestions for Authors

1. give the numerical finding of all experiments in the abstract

2. give the novelty of work with respect to previously reported work

3. check the image 2 scale

4. What is the scattering angle used for particle size analysis

5. How to check the thickness

6. explain the effect of zeta potential on cutaneous thickness

7. The experimental part requires the full description

8. give the particle size image of all suspension

9. can explain the sedimentation of particles over the dermal penetration

10 image are not clear

11. conclusion require rewrite with the result and proper justification support 

11. extensive English revision is required

Comments on the Quality of English Language

English require to editing

Author Response

We thank the reviewer for the time and the efforts to review our manuscript. We carefully addressed all points raised and revised the manuscript accordingly. Changes in the manuscript are highlighted in bright blue. In the document attached, we give detailed answers to all reviewer’s comments.

Reviewer 2 Report

Comments and Suggestions for Authors

1- Some figures lack statistical analysis

2- Sometimes the author use nanocrystals and other times they use nanosuspension. I assume that they are the same so if this is true, the authors should choose only one most probably it should be nanosuspension.

3- Loading capacity, entrapment efficiency and release studies should be carried out.

They ate crucial to he able to compare between bulk and nano

Comments on the Quality of English Language

Minor editing just typo mistakes

Author Response

(The authors gave the same response as above.)

Reviewer 3 Report

Comments and Suggestions for Authors

Dear Authors,

This is an ok article/work, however, there are a few things that can be improved to increase scope.

1.    Minor typos in the text

2.    The conclusions part in the abstract is a bit long and convoluted

3.    Recent literature on skin and skin products can be added to improve the introduction section. Very limited. Suggestions - https://doi.org/10.1016/j.addr.2022.114293

4.     Also, recent literature on curcumin could be useful; https://doi.org/10.3390/pharmaceutics11120639

5.    Few references have different formats, please check and make them consistent.

6.    Page 5 section 2.2.3(Lines 185-212) doesn’t fully support the science. The author claims changes in SC size based on hydration. However, no histology images are provided. Only numbers are stated in results. Please include figures and formulate a table. Please see https://doi.org/10.1208/s12248-016-9984-0 for a microscopic method for skin hydration assessment.

7.    Figure 4 is not a great representation. Actula data would have been great.

8.    Paragraph line starting 302 suggests “The stronger increase in SC thickness for the bulk material, when compared to the 302 nanomaterial, can be explained by the fast sedimentation of the large particles (Figure 4). 303 The large particles (> 10 µm) settle quickly after dermal application and thus come quickly 304 into contact with the skin, where they create a locally high concentration gradient of 305 dissolved curcumin.” All of these can be easily demonstrated using imaging techniques. The manuscript could have been much stronger.

Comments on the Quality of English Language

English is fine. The flow and structure of this paper is clear and easy to understand. 

Author Response

(The authors gave the same response as above.)

Reviewer 4 Report

Comments and Suggestions for Authors

Curcumin bulk suspensions and nanosuspensions with different concentrations of curcumin were produced and characterized regarding, size and zeta potential. Furthermore, the influence of the formulations on the stratum corneum thickness and the dermal penetration was investigated using the ex-vivo porcine ear model. The Stratum corneum thickness was also determined and compared to untreated skin and to skin treated with the particle free vehicle (surfactant solution). My suggestion is to publish this work in this journal after minor revisions according to the following specific comments:

 1.      For the elaboration of curcumin suspensions and nanosuspensions the authors only use Curcumin, TPGS, and Purified water. Did you use for samples preparation anything else (additives and/or excipients…) besides API, TPGS and purified water? Furthermore, a more detailed description of the methodological preparation of samples is required (section 2.1).

2.      Why does the authors use for elaboration process TPGS as surfactant for bulk and nanosuspensions?

3.      A more detailed description of the physicochemical characteristics of curcumin would be appreciated by the authors (solubility, photostability, colour, melting point,,,,…)

4.      Describe correctly line 125 “The In the first………”

 5.      The stability of curcumin is pH-dependent, being reasonably stable at pH 1-6 and unstable at pH >7. Do the authors consider necessary to measure pH of samples developed?

6.      Generally, skin application products require flow and viscosity measurements.

Have the authors considered determining the viscosity of suspensions and nanosuspensions of curcumin? Do the authors consider necessary to do it?

7.      Did the authors consider the stability of formulations developed? Do the authors perform short-term and/or large-term stability studies about formulations developed? Explain it.

8.      Skin irritation studies are commonly performed for topically applied products. Do the authors consider it necessary to do these tests for the suspensions and nanosuspensions developed?

9.      In relation to biopharmaceutical studies, the ex vivo model using ear pig study in pigs is appreciated. Have the authors considered doing curcumin in vitro release studies?

10.  In relation to biopharmaceutical studies, the ex vivo study in pigs is appreciated. MGV method seems to be a semiquantitative method. Have the authors considered quantifying curcumin in the study through another methodological analyses?

Author Response

(The authors gave the same response as above.)

Reviewer 5 Report

Comments and Suggestions for Authors

The paper 'Influence of dose, particle size & concentration on dermal penetration efficacy of curcumin' by Em-on Chaiprateep , Sabrina Wiemann , Ralph W Eckert , Christian S. Raab , Soma Sengupta , Cornelia M. Keck submitted to the Special Issue Nanoparticles and Microparticles in Drug Delivery is interesting and could be published in your famous journal after slight improvements. The article is part of a group of interesting studies on the influence of the particles size of hardly soluble active substance, on their penetration into the skin after topical application. This is important because it may be helpful in properly designing a drug that would have adequate bioavailability. The obtained results are interesting, the results are well documented, well explained based on the literature and in my opinion this paper should be interesting for readers. 

However, some improvements are required:

line 48 references cited [15-26] should be reduced, as 9 of 12 are self-citations

line 120 should be corrected

figures: please explain what vertical line means: is it standard deviation or standard error

Figure 3 - please explain abbreviation SCT, is it correct significance; between untreated skin and TPGS infinite dose p <0.01?

Moreover, could you please explainthe authors contribution as only 3 authors are mentioned, what about the remaining three?

Despite these minor doubts the paper could be published in your famous journal

Author Response

(The authors gave the same response as above.)

Round 2

Reviewer 1 Report

Comments and Suggestions for Authors

Accept